# National Prevalence of Caprine Prion Protein Genetic Variability at Codons 146, 211, and 222 in Goat Herds in the United States

**DOI:** 10.3390/vetsci11010013

**Published:** 2023-12-27

**Authors:** Mohamed Zeineldin, Heather Cox-Struble, Patrick Camp, David Farrell, Randy Pritchard, Tyler C. Thacker, Kimberly Lehman

**Affiliations:** 1National Veterinary Services Laboratories, Veterinary Services, Animal and Plant Health Inspection Service, United States Department of Agriculture, Ames, IA 50010, USA; heather.coxstruble@usda.gov (H.C.-S.); patrick.m.camp@usda.gov (P.C.); david.t.farrell@usda.gov (D.F.); 2Department of Animal Medicine, College of Veterinary Medicine, Benha University, Benha 13511, Egypt; 3Strategy and Policy, Veterinary Services, Animal and Plant Health Inspection Service, United States Department of Agriculture, Fort Collins, CO 80521, USA

**Keywords:** goats, genotyping, *PRNP*, prion, scrapie, United States

## Abstract

**Simple Summary:**

Scrapie is a neurodegenerative disease that affects sheep and goats. This study aimed to estimate the prevalence of caprine PRNP genetic variability in goat populations across the United States. The homozygous 146NN, 211RR, and 222QQ alleles, which are associated with scrapie susceptibility, were found to be the most prevalent among caprine *PRNP* alleles. The prevalence of different genotypes varied across regions. The 222QK allele was most frequently found in the east and southwest regions, whereas 211RQ was more common in the Midwest and east regions. The 146NS genotype had the highest prevalence in the northwest and southwest regions. These findings provide insights into the prevalence of caprine *PRNP* genetic variability in the United States and its potential application for scrapie susceptibility management strategies.

**Abstract:**

Scrapie is a neurodegenerative disease that impacts sheep and goats, characterized by gradual and progressive changes in neurological function. Recent research shows that the scrapie incubation period is significantly influenced by specific variations in amino acids within the prion protein gene (*PRNP*). The objective of this study was to estimate the national prevalence of caprine *PRNP* genetic variability at codons 146, 211, and 222 in goat populations across the United States. A total of 3052 blood, ear tissue, and brain tissue samples were collected from goats from 50 states. The participating states were categorized into four Veterinary Service (VS) district regions. The samples underwent DNA extraction, and the *PRNP* variants corresponding to codons 146, 211, and 222 were amplified and sequenced. The analysis of *PRNP* variants, when compared to the *PRNP* reference sequence, revealed seven alleles in twelve genotypes. The homozygous 146NN, 211RR, and 222QQ alleles, which have been linked to an increased risk of scrapie, were found to be the most prevalent among all the goats. The heterozygous 222QK, 211RQ, 146SD, 146ND, and 146NS alleles and the homozygous 222KK, 146SS, and 146DD alleles, known to be associated with reduced scrapie susceptibility and a prolonged incubation period after experimental challenge, were found in 1.098% (222QK), 2.33% (211RQ), 0.58% (146SD), 3.13% (146ND), 20.68% (146NS), 0.005% (222KK), 3.31% (146SS), and 0.67% (146DD) of goats, respectively. The 222QK allele was found most frequently in goats tested from the east (VS District 1, 1.59%) and southwest (VS District 4, 1.08%) regions, whereas the 211RQ allele was found most often in goats tested from the Midwest (VS District 2, 8.03%) and east (VS District 1, 6.53%) regions. The 146NS allele was found most frequently in goats tested from the northwest (VS District 3, 29.02%) and southwest (VS District 4, 20.69%) regions. Our results showed that the prevalence of less susceptible genotypes at *PRNP* codon 146 may be sufficient to use genetic susceptibility testing in some herds. This may reduce the number of goats removed as part of a herd clean-up plan and may promote the selective breeding goats for less susceptible alleles in high-risk herds at the national level.

## 1. Introduction

Scrapie is a naturally occurring, progressive neurodegenerative disease that impacts both sheep and goats [1]. It is classified as one of the transmissible spongiform encephalopathies (TSEs) or prion diseases, which have been observed to impact various mammalian species, including humans [2,3]. The classical form of scrapie arises from the misfolding of the normal cellular prion protein (PrPC) encoded by the host, resulting in the formation of an abnormal isoform (PrPSc) that accumulates in the brain and peripheral tissues [2,4,5]. The disease progresses slowly, with affected animals showing symptoms such as behavioral changes, tremors, and loss of coordination [1]. Export restrictions due to scrapie in the U.S. have limited export markets for the small ruminant industry, resulting in annual economic losses of approximately $10 to $20 million [6]. 

Amino acid variations encoded in the prion protein gene (*PRNP*) have been shown to influence susceptibility to natural scrapie and the length of the incubation period in small ruminants [7]. In sheep, specific polymorphisms within the *PRNP* open reading frame, particularly the amino acid variations at codons 136, 154, and 171 (p. Ala136Val, p. Arg154His, p. Glu171Arg, and p. Glu171His) are associated with either resistance or susceptibility to scrapie [8,9]. Similarly, the *PRNP* gene in goats exhibits genetic variability, and certain variations have been linked to resistance against natural scrapie [10,11]. Understanding the genetic basis of scrapie susceptibility in small ruminants is crucial for implementing effective control and prevention strategies, thereby reducing the prevalence of the disease [7]. Additionally, genetic testing can be performed to identify susceptible individuals and implement appropriate management practices to minimize the risk of scrapie transmission.

Recent studies have indicated that specific polymorphic variants of *PRNP* at certain codons have been linked to extended incubation periods and reduced susceptibility to classical scrapie in goats [12,13,14]. These variants include amino acid variations at codon 142; p. Met142Iso, codon 146; p. Ser146Asp and p. Asp146Ser, codon 154; p. His154Arg, codon 211; p. Glu211Arg, and codon 222; p. Lys222Glu. 

According to a comprehensive survey we conducted on the genetic variability of *PRNP* in scrapie-free goats from 24 states in the United States, it was revealed that certain goats possess specific *PRNP* sequences that are linked to extended incubation periods and resistance to scrapie. Considering the distribution of resistant genotypes across goat operations, it was found that approximately 72.8% of goat operations had at least one of the genotypes associated with reduced susceptibility to scrapie. The specific genotypes identified include 146NS, 146ND, 146SD, and 222QK [15].

The objective of this study was to expand and validate the findings of our previous research by conducting a statistically validated study to ensure representation of the entire national goat herd. This study included goats from all 50 states. These data can be used to evaluate the potential for establishing breeding programs aimed at selecting goats with less susceptible alleles, thereby aiding in the elimination and prevention of scrapie within the national goat herd.

## 2. Materials and Methods

### 2.1. Study Design and Biological Sampling

At the inception of the project, it was determined that the target sample size for the study would be 3000 samples divided proportionally between all 50 states based on the breeding goat population in each state. Participating states were categorized into four VS district regions: VS District 1 (Alabama, Connecticut, Delaware, Florida, Georgia, Maine, Maryland, Massachusetts, New Hampshire, New Jersey, New York, North Carolina, Pennsylvania, Rhode Island, South Carolina, Tennessee, Vermont, Virginia, and West Virginia), VS District 2 (Illinois, Indiana, Iowa, Kentucky, Michigan, Minnesota, North Dakota, Ohio, South Dakota, and Wisconsin), VS District 3 (Alaska, Arizona, California, Colorado, Hawaii, Idaho, Montana, Nevada, New Mexico, Oregon, Utah, Washington, and Wyoming), and VS District 4 (Arkansas, Kansas, Louisiana, Mississippi, Missouri, Nebraska, Oklahoma, and Texas) (Figure 1). The National Agricultural Statistics Service (NASS) maintains and reports an annual estimate of the size of the national breeding goat population by state. The 2019 report was used to assign the number of samples to be collected from each state. States contributed samples from 0.24% of their breeding herd, with the exception of Texas, which, due to its very large goat population, contributed only 0.12%. Sample collectors for this study were asked to submit approximately ½ inch by ½ inch samples of fresh brain tissue or ear tissue from the Regulatory Scrapie Slaughter Surveillance (RSSS) [16]. In the states that encountered challenges in obtaining sufficient samples through RSSS, samples were collected from goat necropsies or from live-animal on-farm goats using a 5 mL sample of whole blood in an ethylenediaminetetraacetic acid tube until the predetermined proportion of their state scrapie sampling minimum was met. The total number of samples collected per VS district is shown in Appendix A. The age of the tested goats ranged from 18 months up to 72 months of age. A total of 3052 samples were collected, representing all participating states, with a maximum of 30 samples obtained from the same herd. The samples were collected and shipped to the National Veterinary Services Laboratories (NVSLs) for the analysis of *PRNP* polymorphisms at codons 146, 21, and 222.

### 2.2. PCR Amplification and Sequencing

The DNA extraction process from blood, ear, and brain tissue samples was carried out using the MagMax Core bead extraction kit (Thermo Fisher Scientific, Waltham, MA, USA). This extraction was employed in accordance with the manufacturer’s instructions. Subsequently, the concentration of the extracted DNA was determined using the Qubit High Sensitivity DNA standard (Agilent Technologies, Santa Clara, CA, USA). The DNA samples were then preserved by freezing at a temperature of −20 °C until the sequencing procedure was performed. To amplify the open reading frame of *PRNP*, PCR was performed using 10µM of PrP_uniF (5′-AGTCAGTGGAACAAGCCCAG-3′) and PrP_uniR (5′-TGAGGAGGATCACAGGAGGG-3′), as described previously [15]. The PCR mixture comprised of 2 μL of extracted DNA, 1.5 μL of 25 mM MgCl2, 1.25 μL of DMSO, 2 μL of 2.50 mM dNTPs, 1 μL of 5 µM forward primer, 1 μL of 5 µM reverse primers, 0.1 μL of FastStart Taq (5U/µL), and 2.5 μL of 10X AmpliTaqGold^®^ PCR Buffer. The PCR protocol involved an initial denaturation step at 95 °C for 5 min, followed by 35 cycles of denaturation at 94 °C for 30 s, annealing at 56 °C for 30 s, and extension at 72 °C for 30 s. This was followed by an extension incubation at 72 °C for 10 min. After amplification, the DNA underwent purification and cleaning prior to the sequencing reaction using ExoSAP-IT (Thermo Scientific™, Waltham, MA, USA). The final purified PCR products from both strands were subjected to sequencing using the Applied Biosystems ABI 3500xL genetic analyzer (Applied Biosystems, Foster City, CA, USA). For the sequencing reaction, a dye-terminating sequence method was applied, following the manufacturer’s instructions. 

### 2.3. Bioinformatic and Statistical Analysis

The analysis of specific polymorphisms within the *PRNP* open reading frame, particularly the predicted amino acid variations at codons 146, 211, and 222, was conducted using Geneious Prime software (Version 2020.2). The DNA sequences were compared and aligned with the reference sequence of the Capra hircus *PrP* gene (GenBank: HM038415.1). The *PRNP* polymorphisms corresponding to codons 146, 211, and 222 were evaluated. The weighted descriptive estimation of genotypic and allelic proportions was calculated across all goats and participating states using SAS-JMP software (version 13, SAS Institute, 2012). To determine the allele frequency, the genotype prevalence was prorated based on the percentage of the national population of goats in each state.

To compare the variations in genotypic frequencies among the VS districts, chi-square tests were employed using SAS-JMP software (version 13, SAS Institute, 2012). Additionally, testing the homogeneity of variance among the groups defined by VS districts was conducted using Levene’s test for homogeneity using SAS-JMP software (version 13, SAS Institute, 2012). All the sequence data acquired during the present study were uploaded to the sequence read archive on the NCBI website, and a bio-project accession number of PRJNA1015183 was assigned to access the data.

## 3. Results and Discussion

Scrapie is a naturally occurring, fatal neurodegenerative disease that impacts small ruminants, including sheep and goats [1]. It has been established that both scrapie susceptibility and the length of the incubation period in goats are significantly influenced by the polymorphisms and genetic variability present in the PRNP gene [17]. Understanding the distribution and prevalence of different PRNP genotypes can provide valuable insights into the potential resistance or susceptibility of goats to Scrapie. Although the occurrence of scrapie in goats is relatively low, it is important to acknowledge that goats can still act as a reservoir for the disease [10,11]. Consequently, it becomes crucial to identify the genotypes within the goat population that exhibit resistance to scrapie. In our previous study, we conducted an analysis of PRNP genetic variability in goats spanning 24 states within the United States. Our findings revealed that, among the goat population, approximately 33.8% exhibited one or more of the less susceptible genotypes, namely 146NS, 146ND, 146SS, 146DD, 146SD, and 222QK. This genotype prevalence was observed across approximately 72.8% of goat operations [15]. These findings shed light on the prevalence of these genotypes in scrapie-disease-free goats within the studied states. To expand upon and further validate the findings of our first study, we conducted this study, encompassing all 50 states of the United States. Additionally, we included a wider range of sample types, such as blood, ear tissue, and brain tissue. 

In this study, twelve PRNP polymorphisms at codons 146, 211, and 222 were identified. The homozygotes 146NN, 211NN, and 222QQ, which are associated with scrapie susceptibility, exhibited a high prevalence across the United States goat population. In line with other studies, our study showed a prevalence of predominantly susceptible homozygous genotypes compared to heterozygous genotypes [15,18]. These data suggest that there is no natural selection for scrapie resistant genotypes [15,18].

At codon 146, 2277 (71.62%) and 623 (20.68%) goats carried the NN and NS alleles, respectively. The SS and DD homozygotes occurred in 97 (3.31%) and 5 (0.67%) goats, respectively (Appendix A). The ND and SD heterozygotes were detected in 42 (3.13%) and 8 (0.58%) goats, respectively. When comparing the genotype frequencies at codon 146, we observed that the homozygote 146NN, which is associated with scrapie susceptibility, displayed significant differences across the VS districts (P > 0.001, Table 1). The 146NN genotype ranged from 53.85% to 100% across all the states (Appendix A). The 146NN genotype was present in (83.93%) and (77.38%) of goats from VS District 2 and VS District 1, respectively (Table 1). The 146NS genotype, which is associated with scrapie resistance, was observed in all the states except Alaska, Delaware, New Hampshire, and Rhode Island. There was a higher frequency of the 146NS genotype in goats from VS District 3 (29.02%) and VS District 4 (20.68%) (Table 1). The other less susceptible variant 146DD was detected in five goats (0.83%) from TX (Appendix A). The 146SD was only detected in 0.49% and 0.164% of goats from VS District 4 and VS District 2, respectively (Table 1). The 146ND genotype was detected in higher frequency across goats from VS District 4 (2.74%) and VS District 1 (1.11%) (Table 1). 

At codon 211, a total of 2892 (97.34%) goats were carrying RR211. The less susceptible QQ211 homozygote and the less susceptible 211RQ heterozygote was present in 13 (0.33%%) and 147 (2.33%) goats, respectively (Appendix A). Codon 211 exhibited significant differences between the VS districts (P > 0.001, Table 2), with a high percentage of the 211RQ genotype in VS District 2 (8.03%) and VS District 1 (6.53%) (Table 2). The less susceptible variant QQ211 was only detected in 0.82%, 0.79%, and 0.25% of goats from VS District 2, VS District 1, and VS District 4, respectively (Table 2).

At codon 222, the Q allele was found to be the most common among all goats, with a total of 3020 (98.97%) goats displaying 222QQ (Appendix A). The less susceptible genotypes, namely the 222QK heterozygote and 222KK homozygote, were observed in 30 (1.08%) and 2 (0.005%) goats, respectively (Appendix A). A comparison of the genotype frequencies at codon 222 revealed significant differences among the VS districts (P > 0.001, Table 3). The less susceptible 222QK heterozygote was relatively rare and was detected in highest prevalence in VS District 1 (1.59%) and VS District 4 (1.08%) (Table 3). In line with our previous study, the homozygous 222KK genotype was only detected in two goats from WI and NY [15]. The potential protective role of the K222 allele against scrapie has been extensively examined in various experimental studies. These investigations have demonstrated that 222QK goats, possessing the K222 allele, remained free of scrapie for an average duration of approximately 6.8 years following scrapie inoculation [13,14]. However, due to the relatively low frequency of the less susceptible K222 allele in goats within the United States, establishing a breeding program for scrapie resistance based on this allele would pose significant challenges. Such a program could only be feasibly implemented in breeds in which the K222 allele already exists. However, the higher percentage of goats in this study carrying the 146NS genotype presents the greatest opportunity for establishing a breeding program to reduce the risk of scrapie in the United States. By selectively breeding goats that carry the 146NS genotype, we can effectively reduce the susceptibility of the goat population to scrapie. This targeted approach minimizes the risk of disease transmission and helps safeguard the overall health and welfare of goats in the United States.

This study demonstrated that the goat population in the United States exhibits significant genetic variability in PRNP at codons 146, 211, and 222. However, due to the descriptive nature of this study and the absence of case–control data, it is not possible to establish a definitive association between the polymorphism in the *PRNP* genotype and scrapie susceptibility. Among the 12 PRNP coding regions examined in this study, the 211RQ genotype has previously been identified in various goat breeds raised in the United States, including Oberhasli, Alpine, Saanen, LaMancha, Spanish, and Sable goats. This genotype has been linked to an extended scrapie incubation period and partial resistance to scrapie following experimental infection [15,19]. Additionally, the low prevalence of the 222K allele detected in this study aligns with previous research in the United States, which also reported a low frequency of goats carrying this allele [18,20]. These findings highlight the importance of conducting further research and genetic assessments to gain a deeper understanding of the distribution of specific genotypes and alleles associated with scrapie resistance in different goat breeds.

## 4. Conclusions

In conclusion, the findings of this study reveal a high frequency of 146NS, a genotype associated with decreased susceptibility to scrapie and extended incubation periods. These results suggest the potential for implementing comprehensive breeding programs aimed at enhancing scrapie resistance within the goat population in the United States. Furthermore, this study indicates that there are many goats with reduced susceptibility alleles at codon 146, indicating the potential practicality of reducing the number of goats removed as part of a flock clean-up plan. It also encourages the selective breeding of goats possessing less susceptible alleles in high-risk herds at a national level. However, it is important to note that the present study did not specifically evaluate the association between genotypic variability and different goat breeds in the four districts. Although our previous studies have explored genotypic variations among various goat breeds in the United States, this particular dataset does not include breed-specific information. Therefore, the analysis of genotypic variability in relation to specific goat breeds within the four districts is beyond the scope of the current study. Future research incorporating breed-specific data would be valuable in further elucidating the relationship between genotypic variability and different goat breeds within these districts.

## Figures and Tables

**Figure 1 vetsci-11-00013-f001:**
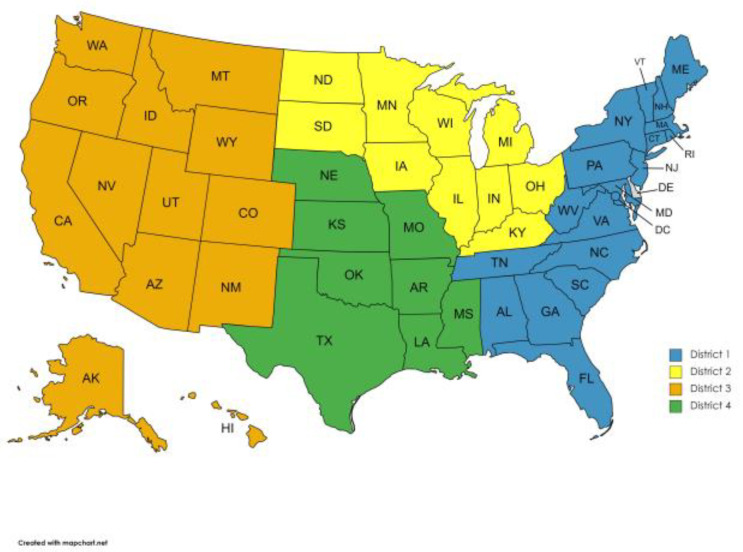
Map of the States and VS district regions.

**Table 1 vetsci-11-00013-t001:** Total number and percentage of goats by caprine *PRNP* genotype at codon 146 by VS district.

VS District	146DD		146SD		146ND		146NN		146NS		146SS	
	Number of Goats	% of Total	Number of Goats	% of Total	Number of Goats	% of Total	Number of Goats	% of Total	Number of Goats	% of Total	Number of Goats	% of Total
VS District_1	0	0.00%	1	0.15%	7	1.11%	486	77.38%	111	17.67%	23	3.66%
VS District_2	0	0.00%	1	0.16%	0	0.00%	512	83.93%	86	14.09%	11	1.81%
VS District_3	0	0.00%	0	0.00%	2	0.32%	411	67.37%	177	29.01%	20	3.27%
VS District_4	5	0.41%	6	0.49%	33	2.74%	868	72.09%	249	20.68%	43	3.57%

**Table 2 vetsci-11-00013-t002:** Total number and percentage of goats by caprine *PRNP* genotype at codon 211 by VS district.

VS District	QQ211		211RQ		RR211	
	Number of Goats	% of Total	Number of Goats	% of Total	Number of Goats	% of Total
VS District_1	5	0.79%	41	6.52%	582	92.67%
VS District_2	5	0.82%	49	8.03%	556	91.14%
VS District_3	0	0.00%	20	3.27%	590	96.72%
VS District_4	3	0.24%	37	3.07%	1164	96.67%

**Table 3 vetsci-11-00013-t003:** Total number and percentage of goats by caprine *PRNP* genotype at codon 222 by VS district.

VS District	222KK		222QK		222QQ	
	Number of Goats	% of Total	Number of Goats	% of Total	Number of Goats	% of Total
VS District_1	1	0.15%	10	1.59%	617	98.25%
VS District_2	1	0.16%	6	0.98%	603	98.85%
VS District_3	0	0.00%	1	0.16%	609	99.84%
VS District_4	0	0.00%	13	1.08%	1191	98.92%

## Data Availability

All sequence data obtained during the present study have been submitted to the NCBI Sequence Read Archive under the Bio-Project accession number PRJNA1015183.

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
