# Peer review of "National Prevalence of Caprine Prion Protein Genetic Variability at Codons 146, 211, and 222 in Goat Herds in the United States"

_vetsci, 2023, doi:10.3390/vetsci11010013_

Round 1

Reviewer 1 Report

Comments and Suggestions for Authors

This paper describes the PRNP genetic variability in scrapie disease-free goats from 50 States from the US. Resulting data can be used to evaluate the potential for establishing breeding programs to select for goats with alleles less susceptible to scrapie. I would suggest two major revisions and some further minor revision.

Major revision

Page 3, line 131 - 143: I noticed that there is no mention of conducting the Hardy-Weinberg equilibrium test in the analyzed population. The Hardy-Weinberg equilibrium is crucial, especially when investigating genotype.

This test would help to ensure that the population under study is not affected by sampling errors, which could significantly impact the validity of the results. It provides a valuable baseline for understanding whether the observed genotype frequencies match the expected frequencies in a population that is not evolving.

Page 4, Line 171-173; Page 5, line 188-190 and 199-202: it would be interesting to understand if the genotypic variability observed in the different Districts is associated with different breeds raised in the four Districts.

Minor revision

Page 2, line 82-83:”the entire national sheep herd was represented in the study. This study included sheep from all 50 states” the authors refer to the “sheep” rather than the “goats”, if it's a mistake please consider to change the sentence.

Page 3, Line 119-123: it would be useful to specify the concentration of the DNA and reagents used in the PCR reaction (e.g. primers). Indeed, authors refer to a previous publication, however the concentration of primers PrP_uniF and PrP_uniR are not in the references (n. 15).

Page 3, Line 127-130: the primers used in the sequencing reaction and their concentration are not reported. Please take into account the possibility to include that information.

Entire Manuscript: verify the correspondence between the percentage values shown in the text and those in the tables (e.g. Page 4, Line 175: The NN146 genotype presented in 174 (83.69%)”, in the Table 1 is report 83.93%; Line 182: “VS district 4(2.71%)” in the Table 1 is report 2.74%.

Entire Manuscript: use the same decimals after the decimal point (e.g. Page 4, Line 178: “VS district 3 (29.016%) and VS district 4 (20.68%)”

Page 4, Table 1: Harmonize the way the genotype is written (e.g. Page 4, Table 1: the genotype at codon 146 is written as DS while throughout the text it is written as SD)

Page 5, Line 187-188 and 197-198: please verify the calculations (e.g. line 197-198 I tried to do some calculations, if I am not wrong, the percentage relating to QK 222 is 0.98% and not 1.08%)

Author Response

Reviewer 1.

This paper describes the PRNP genetic variability in scrapie disease-free goats from 50 States from the US. Resulting data can be used to evaluate the potential for establishing breeding programs to select for goats with alleles less susceptible to scrapie. I would suggest two major revisions and some further minor revision.

Response:

The authors sincerely appreciate your in-depth review of our manuscript. We recognize the validity of your comments and have revised the manuscript in line with your recommendations.

Major revision

Comment 1. Page 3, line 131 - 143: I noticed that there is no mention of conducting the Hardy-Weinberg equilibrium test in the analysed population. The Hardy-Weinberg equilibrium is crucial, especially when investigating genotype.

This test would help to ensure that the population under study is not affected by sampling errors, which could significantly impact the validity of the results. It provides a valuable baseline for understanding whether the observed genotype frequencies match the expected frequencies in a population that is not evolving.

Response. Thank you for the suggestion. We assessed the data for homogeneity of variance among groups defined by VS districts using similar test “Levene’s test for homogeneity” using SAS-JMP software (version 13, SAS Institute, 2012). Line 159 to 162.

Comment 1.

Page 4, Line 171-173; Page 5, line 188-190 and 199-202: it would be interesting to understand if the genotypic variability observed in the different Districts is associated with different breeds raised in the four Districts.

Response: While we agree that it would be interesting to investigate the association between genotypic variability and different goat breeds in the four Districts, we would like to clarify that the current dataset did not specifically evaluate the goat breeds. However, we have conducted previous studies that compared the genotype among various goat breeds in the United States. These prior investigations provide valuable insights into the genotypic variability among different breeds. We included this as a study limitation for the current study. Line 257 to line 265.

Minor revision

Page 2, line 82-83:”the entire national sheep herd was represented in the study. This study included sheep from all 50 states” the authors refer to the “sheep” rather than the “goats”, if it's a mistake please consider to change the sentence.

Response: We corrected it.

Page 3, Line 119-123: it would be useful to specify the concentration of the DNA and reagents used in the PCR reaction (e.g. primers). Indeed, authors refer to a previous publication, however the concentration of primers PrP_uniF and PrP_uniR are not in the references (n. 15).

Response: We added more information in the methods section. Line 138 to 145.

Page 3, Line 127-130: the primers used in the sequencing reaction and their concentration are not reported. Please take into account the possibility to include that information.

Response: We added more information in the methods section. Line 136 to line 144.

Entire Manuscript: verify the correspondence between the percentage values shown in the text and those in the tables (e.g. Page 4, Line 175: “The NN146 genotype presented in 174 (83.69%)”, in the Table 1 is report 83.93%; Line 182: “VS district 4 (2.71%)” in the Table 1 is report 2.74%.

Response: We have checked that and corrected it.

Entire Manuscript: use the same decimals after the decimal point (e.g. Page 4, Line 178: “VS district 3 (29.016%) and VS district 4 (20.68%)”

Response: We have checked that and corrected it.

Page 4, Table 1: Harmonize the way the genotype is written (e.g. Page 4, Table 1: the genotype at codon 146 is written as DS while throughout the text it is written as SD).

Response: We have checked that and corrected as suggested.

Page 5, Line 187-188 and 197-198: please verify the calculations (e.g. line 197-198 I tried to do some calculations, if I am not wrong, the percentage relating to QK 222 is 0.98% and not 1.08%)’

Response: We have checked that, and we calculated again as 1.08 (13/1204 ‘which is the total number of animals samples in VS district 4”).

Reviewer 2 Report

Comments and Suggestions for Authors

Dear authors,

you prepared a well-written manuscript that is easy to understand and very interesting for the reader. I have no major concerns about this study. I only have some suggestions how you could improve overall quality.

Line 75: Could you briefly describe the different incubation periods. What are time spans and effects?

Supp. Table 2: Could you prepare a graphical map of the USA highlighting the different genotypes? That would make it much easier for the reader to get a visual impression and shoud be included in the manuscript.

Overall: As a goat is not a goat – do you have data available to distinguish between different breeds? Is there any breed that has a higher prevalence of the genotypes described?

Author Response

Reviewer 2.

you prepared a well-written manuscript that is easy to understand and very interesting for the reader. I have no major concerns about this study. I only have some suggestions how you could improve overall quality.

Response:

The authors sincerely appreciate your in-depth review of our manuscript. We recognize the validity of your comments and have revised the manuscript in line with your recommendations.

Supp. Table 2: Could you prepare a graphical map of the USA highlighting the different genotypes? That would make it much easier for the reader to get a visual impression and should be included in the manuscript.

Response: We added the map (Figure 1).

Overall: As a goat is not a goat – do you have data available to distinguish between different breeds? Is there any breed that has a higher prevalence of the genotypes described?

Response: While we agree that it would be interesting to investigate the association between genotypic variability and different goat breeds in the four Districts, we would like to clarify that the current dataset did not specifically evaluate the goat breeds. However, we have conducted previous studies that compared the genotype among various goat breeds in the United States. These prior investigations provide valuable insights into the genotypic variability among different breeds. We included this as a study limitation for the current study. Line 257 to line 265.

Reviewer 3 Report

Comments and Suggestions for Authors

Zeineldin and colleagues carried out a study analyzing the distribution of scrapie-risk associated genotypes in American goats.

The study is overall of interest but has a major limitation in ignoring the breed evaluation. In my opinion, that is a factor even more significant than the geographical distribution. I find bizarre that the authors skipped it outright.

Additionally, the data presentation requires significant improvements, overall confusing genotype and protein amino acids.

31: please here and later in the paper indicate the accession number and perhaps use the correct nomenclature “(p. XXX”). The exact protein and positions must be immediately identifiable. Later one can identify the AA combination in a more practical manner, but the correct one should be used at least once.

35: “predominant” is too generic.

37: Lack of precision. Please use the correct nomenclature for the AAs, and don’t call an aminoacid combination a “genotype”.

56-57: unclear if specific or generic here, protein-wise.

64-65: see 31

72-75: https://varnomen.hgvs.org/

77-79: unclear percentages. Please rephrase.

99: Figure?

106-108: Could this be elaborated, perhaps in the form of a table?

132-133: Again, there is confusion between DNA polymorphisms and AA sequence.

162: Again, this is perhaps clear, but incorrect. An AA combination is not a genotype.

227: To me, this is a major limitation. 

Author Response

Reviewer 3.

Zeineldin and colleagues carried out a study analyzing the distribution of scrapie-risk associated genotypes in American goats.

The study is overall of interest but has a major limitation in ignoring the breed evaluation. In my opinion, that is a factor even more significant than the geographical distribution. I find bizarre that the authors skipped it outright.

Response: The authors sincerely appreciate your in-depth review of our manuscript. We recognize the validity of your comments and have revised the manuscript in line with your recommendations. While we agree that it would be interesting to investigate the association between genotypic variability and different goat breeds in the four Districts, we would like to clarify that the current dataset did not specifically evaluate the goat breeds. However, we have conducted previous studies that compared the genotype among various goat breeds in the United States. These prior investigations provide valuable insights into the genotypic variability among different breeds. We included this as a study limitation for the current study. Line 257 to line 265.

Additionally, the data presentation requires significant improvements, overall confusing genotype and protein amino acids.

Response: We have checked that and modified as suggested.

31: please here and later in the paper indicate the accession number and perhaps use the correct nomenclature “(p. XXX”). The exact protein and positions must be immediately identifiable. Later one can identify the AA combination in a more practical manner, but the correct one should be used at least once.

Response: We have added the correct nomenclature in the first paragraph of the introduction section. Line 63-66 and Line 75 to 78.

35: “predominant” is too generic.

Response: We rephrased it.

37: Lack of precision. Please use the correct nomenclature for the AAs, and don’t call an aminoacid combination a “genotype”.

Response: genotype was removed.

56-57: unclear if specific or generic here, protein-wise.

Response: This is generic to define how the scrapie develop.  

64-65: see 31

Response: We have added the correct nomenclature in this paragraph. The exact protein and positions were identified.

72-75: https://varnomen.hgvs.org/

Response: We have checked that and modified as suggested.

77-79: unclear percentages. Please rephrase.

Response: We rephrased this sentence.  

99: Figure?

Response: We added the map (Figure 1).

106-108: Could this be elaborated, perhaps in the form of a table?

Response:  The total number of samples collected is shown in figure S1.

132-133: Again, there is confusion between DNA polymorphisms and AA sequence.

Response: Response: We rephrased this sentence.  Line 152-153.

162: Again, this is perhaps clear, but incorrect. An AA combination is not a genotype.

Response: genotype was removed.

227: To me, this is a major limitation.

Response: We agree, and we included this as a limitation to the current study. Line 271 to line 279.

Round 2

Reviewer 1 Report

Comments and Suggestions for Authors

Dear Authors,

Your paper describes the PRNP genetic variability in scrapie disease-free goats from 50 States from the US.   These are important results in order to evaluate a possible breeding programs to select for goats with alleles less susceptible to scrapie. I appreciated the changes you made in response to my comments in the first revision.

Best regards

Author Response

Thank you for acknowledging the importance of our results. Your insights have been invaluable in shaping the manuscript, and we are grateful for your constructive input.

Reviewer 3 Report

Comments and Suggestions for Authors

The authors seem to address the concerns raised. For the record, I think the p.Arg154His style is less confusing than the p.(Arg154His) one used here but the latter is still correct, so it's up to the authors.
Did the author cite all the relevant literature concerning the study of these loci? These occur to me in the very least.
https://doi.org/10.1038/s41598-019-51625-8
doi: 10.1371/journal.pone.0198819

Author Response

Comment 1. The authors seem to address the concerns raised. For the record, I think the p.Arg154His style is less confusing than the p.(Arg154His) one used here but the latter is still correct, so it's up to the authors.

Response: 

The authors sincerely appreciate your in-depth review of our manuscript. We have changed this as requested.

Comment 2. Did the author cite all the relevant literature concerning the study of these loci? These occur to me in the very least.
https://doi.org/10.1038/s41598-019-51625-8
doi: 10.1371/journal.pone.0198819

Response: Thank you for the suggestion. we have updated the reference list.